# Universal Sample Coding

**Szymon Kobus**
Dep. of Electrical and Electronic Engineering
Imperial College London
szymon.kobus17@imperial.ac.uk

**Tze-Yang Tung**
Not Diamond
tze-yang@notdiamond.ai

**Deniz Gündüz**
Department of Electrical and Electronic Engineering
Imperial College London
d.gunduz@imperial.ac.uk

## Abstract

In this work, we study the problem of communicating multiple samples from an unknown probability distribution using as few bits as possible. This is a generalization of the channel simulation problem, which has recently found applications and achieved state of the art results in realistic image compression, neural network compression, and communication-efficient federated learning. In this problem, the transmitter wants the receiver to generate multiple independent and identically distributed (i.i.d.) samples from a target distribution $P$, while the transmitter and the receiver have access to independent samples from a reference distribution $Q$. The core idea is to employ channel simulation in multiple rounds while updating the reference distribution $Q$ after each round in order to reduce the KL-divergence between $P$ and $Q$, thereby reducing the communication cost in subsequent rounds. We derive a lower bound on the expected communication cost and construct a practical algorithm that achieves the lower bound up to a multiplicative constant. We then employ this algorithm in communication-efficient federated learning, in which model updates correspond to samples from a distribution, and achieve a 37% reduction in the communication load. To further highlight the potential of sample communication for generative models, we show that the number of bits needed to communicate samples from a large language model can be reduced by up to 16 times, compared to entropy-based data compression.

## 1 Introduction

Let $P$ be a probability distribution known to an encoder, and we would like to obtain a sample $X \sim P$ at a remote decoder - what is the minimal number of bits required to communicate from the encoder to the decoder? The obvious procedure is to sample from $P$ at the encoder, compress it, and transmit over the network. The minimum number of bits necessary for lossless transmission of this sample is bounded by the entropy of distribution $P$, $\mathbf{H}(P)$ bits per sample. However, unlike in classical source coding theory, in our problem, the decoder is interested only in drawing of 'an arbitrary' sample from $P$, and not the particular sample generated at the encoder. If the encoder and decoder have access to samples from another reference distribution $Q$, then it is possible for the encoder to communicate a sample from $P$ using approximately $\mathbf{D}_{\mathrm{KL}}(P\|Q)$ bits instead. Thus, if we are able to sample from a distribution $Q$ that is *close* to $P$, the bandwidth required to communicate a sample from $P$ can be greatly reduced. This is known as 'channel simulation' or 'reverse Shannon coding' (Bennett et al., 2002; Cuff, 2008; Harsha et al., 2010).

38th Conference on Neural Information Processing Systems (NeurIPS 2024).

This result has recently been used in various neural compression problems, such as compression of images and neural networks (Havasi et al., 2019; Flamich et al., 2020; Agustsson and Theis, 2020; Theis et al., 2022), where non-differentiable quantization operation is replaced with reverse channel coding, which provides a differentiable step by utilizing the reparameterization trick. Another recent application of channel simulation is in federated learning (FL). Many communication efficient FL methods require sending a sample from a client-only distribution after local training. Instead of deterministic weights, the models can be parameterized with a probability distribution and the central server samples realizations from each client's updated model to construct the global model update. Recently, it was shown in Isik et al. (2023, 2024) that the overall communication cost of sending the model updates from the clients to the parameter server can be greatly reduced using channel simulation, achieving state-of-the-art results. In each iteration, the client can enable the parameter server to sample from its locally updated distribution while using the global model distribution as the common reference distribution. As the client's local distribution is typically close to the global model, the communication cost, proportional to the KL-divergence between the two, will will extremely low compared to deterministic alternatives. Channel simulation can also help in enabling differential privacy of the communicated models (Triastcyn et al., 2021; Hasircioglu and Gunduz, 2024; Hegazy et al., 2024; Shahmiri et al., 2024) since it allows for shaping the distribution of the noise.

In all the above works, the single-shot version of channel simulation is utilized; that is, a single sample is to be generated at the receiver with the desired distribution. In this work, we are interested in studying the communication cost of sending *multiple* samples from the same distribution. This problem can directly find an application in the FL framework, where the global model is estimated from samples of local models. A procedure to efficiently communicate more samples would allow the parameter server to approximate the global model more accurately, which would reduce the noise and variance in its estimation, and speed up training.

Additionally, we would like to point to a strong conceptual connection between generative models and channel simulation, and propose an avenue of future research. Generative models are one of the most successful examples among the recent advances in machine learning. They include various modalities such as text (Touvron et al., 2023), images (Ho et al., 2020), video, and many more. They are trained to mimic some data distribution $P$ based on samples $X \sim P$ drawn from it, and allow to generate novel samples from the learned distribution. With growing popularity and deployment of generative AI across a wide variety of applications, communicating the outputs of these models (particularly for image, audio and video modalities) will put an increasing burden on the underlying communication network infrastructure. However, instead of generating and then compressing a sample at the cloud server, we can exploit channel simulation to enable the users to locally generate samples at a much lower communication cost. Drawing multiple samples is a common use case for generative models. For instance, in text to image models, such as DALL-E or Midjourney, multiple images are often generated and communicated to the user based on a single prompt to allow the user choose the desired one. Other instances include translation (Lee et al., 2021; Eikema and Aziz, 2022), code generation (Chen et al., 2021) and planning (Yao et al., 2023), which can be improved by discriminating among multiple generated samples.

The contributions of this work are summarized as follows:

- We formulate a novel *universal sample coding problem*, which defines the problem of communicating multiple samples from an arbitrary discrete distribution unknown to the receiver.

- We highlight the relationship between the redundancy in universal source coding and the communication of samples, and show that the optimal redundancy for universal source coding is a lower bound on the communication cost of universal sample coding.

- We propose a coding scheme for the universal sample coding problem, providing an upper bound on its communication cost that is within a multiplicative factor from the lower bound.

- We employ our algorithm in FL achieving a $37\%$ reduction in the communication load compared to the current state-of-the-art communication-efficient FL algorithm.

- We adapt our algorithm for the remote generation problem, where the goal is to sample at a remote user from a generative model hosted on a cloud server, while communicating the least number of bits from the server to the user. We demonstrate the potential benefits of sample communication in this setting through numerical experiments.

Table 1: Rate required to communicate a given/any sample from $P$

|  | matched | mismatched |
| --- | --- | --- |
| source coding | $\mathbf{H}(P)$ | $\mathbf{H}(P) + \mathbf{D}_{\mathrm{KL}}(P\|Q)$ |
| sample communication | 0 | $\mathbf{D}_{\mathrm{KL}}(P\|Q)$ |

## 2 Notation

We use $X^n = (X_1, X_2, \ldots, X_n)$ to denote a sequence of $n$ elements and $\log(\cdot)$ denotes logarithms of base 2. Let $P$ and $Q$ denote discrete probability distributions. The entropy of $P$ is given by $\mathbf{H}(P) \triangleq -\mathbf{E}_{x \sim P}[\log P(x)]$, while the relative entropy from $P$ to $Q$, or the KL-divergence, is defined as $\mathbf{D}_{\mathrm{KL}}(P\|Q) \triangleq \mathbf{E}_{x \sim P}\left[\log \frac{P(x)}{Q(x)}\right]$.

## 3 Background

The problem of communicating samples from a desired distribution is a version of the channel simulation problem, also known as the reverse Shannon theorem, or reverse channel coding. Given a sample $z$ from a probability distribution $P_Z$ at the encoder, channel simulation entails generating sample $x$ from the conditional distribution $P_{X|Z=z}$ at the decoder, using the fewest number of bits. It was posed in Cuff (2008), where the solution was characterized in the asymptotic regime, where an infinite sequence of samples is considered, while earlier results appeared in the quantum communication literature (Bennett et al., 2002; Winter, 2002). The non-asymptotic results were shown in Harsha et al. (2010) using common randomness, and then further refined in Li and Anantharam (2021), where it was shown that to communicate a sample $X$ from distribution $P_{X|Z=z}$ with joint distribution $P_{XZ}$, it is sufficient to transmit

$$I(X;Z) + \log(I(X;Z) + 1) + 4.732 \text{ bits} \tag{1}$$

on average, where

$$I(X;Z) \triangleq \mathop{\mathbf{E}}_{z \sim P_Z}\left[\mathbf{D}_{\mathrm{KL}}(P_{X|Z=z}\|P_X)\right] \tag{2}$$

is the mutual information between $X$ and $Z$. This result is close to optimal as it was shown in Li and El Gamal (2018), i.e., there exist distributions $P_{X,Z}$ where the minimal number of communication required for reverse channel coding is on average

$$I(X;Z) + \log(I(X;Z) + 1) - 1 \text{ bits.} \tag{3}$$

As pointed out and shown in Theis and Yosri (2022), the sample communication bound does not rely on using the exact marginal distribution $P_X$, and still holds for other reference distributions $Q$. In general, to generate a sample at the decoder from distribution $P$, with reference distribution $Q$, it is enough to communicate an average of

$$\mathbf{D}_{\mathrm{KL}}(P\|Q) + \log(\mathbf{D}_{\mathrm{KL}}(P\|Q) + 1) + 4.732 \text{ bits.} \tag{4}$$

While efficient in terms of communication cost, the sample communication method in Li and El Gamal (2018) might be prohibitively expensive computationally. There have been increasing efforts in creating algorithms with reduced complexity by constraining the distributions $P$ and $Q$ (Flamich et al., 2022; Flamich, 2023). An overview of the methods and trade-off between communication, computation, and sample accuracy is provided in Theis and Yosri (2022). The problem of communicating multiple samples from an unknown distribution $P$ was also addressed in Choi and Li (2021), where the authors considered a setting without common randomness and achieved sublinear, yet superlogarithmic, communication rates.

## 4 Communication of samples

### 4.1 Source coding

*Lossless source coding* is the problem of describing the realizations of a random variable with the least number of bits on average. Each outcome $x \in \mathcal{X}$ is assigned a different sequence of bits, called

a *codeword*, with length $l(x)$. The optimal average codeword length is obtained by Huffman coding (Huffman, 1952), where $l(x) \leq \lceil - \log P(x) \rceil$ and

$$\mathbf{H}(P) \leq \mathop{\mathbf{E}}_{x \sim P}[l(x)] \leq \mathbf{H}(P) + 1. \tag{5}$$

For $n$ independent and identically distributed random variables $X^n = (X_1, X_2, \ldots, X_n)$, $X_i \sim P$, it is straightforward to show that using a Huffman code, the expected codeword length per symbol is $\mathbf{H}(P) \leq \frac{1}{n} \mathbf{E}_{x^n \sim P^{\otimes n}}[l(x^n)] \leq \mathbf{H}(P) + \frac{1}{n}$. As $n \to \infty$ this quantity converges to the entropy, which provides a fundamental limit for compression.

We call the code *mismatched* if it is designed for a source distribution of $Q$ while the true source distribution is $P$. Then the expected codeword length is given by:

$$\mathop{\mathbf{E}}_{x \sim P}[l_Q(x)] \leq \mathop{\mathbf{E}}_{x \sim P} \lceil - \log Q(x) \rceil \leq \mathop{\mathbf{E}}_{x \sim P} \left[ - \log Q(x) + 1 + \log \frac{P(x)}{P(x)} \right] \tag{6}$$

$$= - \mathop{\mathbf{E}}_{x \sim P} [\log P(x)] + \mathop{\mathbf{E}}_{x \sim P} \left[ \log \frac{P(x)}{Q(x)} \right] + 1 = \mathbf{H}(P) + \mathbf{D}_{\mathrm{KL}}(P \| Q) + 1. \tag{7}$$

For the $n$-ary case, the rate converges to $\mathbf{H}(P) + \mathbf{D}_{\mathrm{KL}}(P \| Q)$ (Cover and Thomas, 2006), where the KL-divergence is the penalty for using a mismatched source distribution when designing the code.

However, there exist source coding methods called *universal source coding* with an average codeword length that converges to $\mathbf{H}(P)$ for any underlying distribution $P$ not known to the encoder. They can be thought of as empirically estimating (often implicitly) the underlying distribution of the data. The classic and widely used example of such an algorithm is LZ78 (Ziv and Lempel, 1978). For any distribution $P$ the average number of bits needed to compress $n$ samples is $\mathbf{H}(P) + O\left(\frac{1}{\log n}\right)$ (Savari, 1997), which converges much slower than Huffman coding for known distributions. This excess factor for any universal source code $\Gamma$ is known as per letter redundancy, defined as:

$$\mathrm{r}_\Gamma(n) = \sup_{P \in \mathcal{P}^k} \frac{1}{n} \mathop{\mathbf{E}}_{x^n \sim P}[l(x^n)] - \mathbf{H}(P), \tag{8}$$

where $\mathcal{P}_k$ is a $k$-dimensional probability simplex. It was shown in Davisson et al. (1981) that for a $k$-dimensional distribution, the minimal per letter redundancy for any universal source code is:

$$\inf_\Gamma \mathrm{r}_\Gamma(n) = \frac{(k-1) \log n}{2n} + O(n^{-1}). \tag{9}$$

The $\log n$ factor is the penalty for universality of the code (Krichevsky and Trofimov, 1981). A more useful quantity for our analysis will be the total redundancy, defined as $n \, \mathrm{r}_\Gamma$. In the remainder of the paper, when we refer to redundancy, we mean the total redundancy.

## 4.2 Sample communication

As mentioned in Section 3, reverse channel coding involves simulating samples from a joint distribution. *Sample communication* is a similar problem where the aim is to draw samples at the decoder from distribution $P$, which is only known at the encoder, while both the encoder and the decoder have access to samples from a common distribution $Q$. This common randomness – in practice, a seed to initialize a pseudorandom number generator – allows both to draw the same sequence of samples from $Q$. Using any sample communication method with cost specified in Equation (4) a sequence of $n$ independent and identically distributed (i.i.d.) samples $X^n, X_i \sim P$, can be encoded using:

$$\mathbf{D}_{\mathrm{KL}}(P^{\otimes n} \| Q^{\otimes n}) + \log(\mathbf{D}_{\mathrm{KL}}(P^{\otimes n} \| Q^{\otimes n}) + 1) + 4.732$$
$$= n \, \mathbf{D}_{\mathrm{KL}}(P \| Q) + \log(n \, \mathbf{D}_{\mathrm{KL}}(P \| Q) + 1) + 4.732 \tag{10}$$

expected number of bits, and the per sample cost converges to $\mathbf{D}_{\mathrm{KL}}(P \| Q)$ as $n \to \infty$.

Characterizations of the rate of source coding and sample communication in both matched and mismatched cases are presented in Table 1, where we can see that 'moving' to source coding adds $\mathbf{H}(P)$ to the required rate, while using a mismatched distribution $Q$ contributes $\mathbf{D}_{\mathrm{KL}}(P \| Q)$. The natural question to ask is whether the results of universal source coding can be extended to sample communication. Firstly, does the rate approach 0 when $n \to \infty$, and secondly, what is the total communication cost of communicating $n$ samples? In the next section, we show that the answer to the first question is positive, and that the total communication cost is in the same order as the minimal redundancy, and the two quantities share a common lower bound.

---

**Algorithm 1** Universal Sample Coding

---

**Input**: $P, n \geq 1$
**Parameter**: $c > 0$

  1: Send sample $X_1 \sim P$.
  2: Let $i = 2, G(1) = 1$.
  3: **while** $i \leq \lceil \log_{1+c}(n) \rceil + 1$ **do**
  4:     Let $G(i) = \lceil (1+c)^{i-1} \rceil$
  5:     Let $Q_i = \hat{Q}_i(X^{G(i-1)})$ be probability estimated with all previous samples (estimator Lemma 5.2).
  6:     Let $g(i) = G(i) - G(i-1)$.
  7:     Jointly send $g(i)$ samples $X_{G(i-1)+1}^{G(i)} \sim P^{\otimes g(i)}$ using $Q_i^{\otimes g(i)}$ (Eqn. (10)).
  8:     $i = i + 1$
  9: **end while**

---

## 5 Universal sample coding

The core idea is that, after observing $t$ samples, the decoder can estimate $P$, and use this estimate as the reference distribution $Q$. As more samples are transmitted, the KL-divergence between the decoder's estimate and the true distribution $P$ will diminish, which will translate into a lower communication cost for later samples. The proposed scheme consists of multiple rounds; in each round, a batch of samples are sent jointly, then the reference probability $Q$ is updated.

**Theorem 5.1** (Universal sample coding). *There exists a randomized encoding function $f : \mathcal{P}_k \times \mathcal{Z} \to \mathbb{N}$, and a randomized decoding function $g : \mathbb{N} \times \mathcal{Z} \to \mathcal{X}^n$, such that for any $k$-dimensional discrete distribution $P \in \mathcal{P}_k$ over alphabet $\mathcal{X}$ and a random string $Z \in \mathcal{Z} = \{0,1\}^\infty$, such that $g(f(P,Z), Z) = X^n \sim P^{\otimes n}$ and*

$$\mathbf{E}\left[ \mathbf{H}(f(P,Z)|Z) \right] = L(n) \leq V_k(c) \log(n) + o\big(\log(n)\big),$$

*where the expectation is over all random strings Z, and*

$$V_k(c) \triangleq \frac{c}{\ln(1+c)} \left( \frac{k-1}{2} \right) + \frac{\ln\left( \frac{k-1}{2\ln 2} + 1 \right) + 5\ln 2}{\ln(1+c)} + 1 \tag{11}$$

*for any $c > 0$.*

The random string $Z$ is the common randomness shared between the encoder and decoder. The expected number of bits needed to communicate $n$ samples from the any distribution $P$ is similar to the redundancy of universal source coding, with some additional factors and per sample cost $O\left( \frac{\log n}{n} \right) \to 0$ as $n \to \infty$.

### 5.1 Probability estimation

As the samples are drawn independently, their order does not reveal any information about $P$. Thus, any estimator, without loss of optimality, can be based only on the count of each symbol in the observed sequence. One of the canonical estimators is the add-1 or Laplace estimator. As the name suggests, the estimated probability of any symbol is its count in the sequence plus 1, normalized to form a probability distribution. Laplace estimator belongs to a family of add-$\beta$ estimators, which rely on adding a constant $\beta$ to each count. Although no add-$\beta$ estimator achieves the optimal minmax KL-divergence rate (Krichevskiy, 1998), there exists an estimator (Braess and Sauer, 2004) that combines add-$\frac{1}{2}$, add-$\frac{3}{4}$ and add-1 estimators, and achieves the optimal and universal decay of KL-divergence.

**Lemma 5.2.** *(Braess and Sauer, 2004)] Given $n$ i.i.d. samples $X^n$, $X_i \sim P$ from any distribution $P \in \mathcal{P}_k$, there exists an estimator $\hat{Q}(X^n)$ such that:*

$$\underset{X^n \sim P^{\otimes n}}{\mathbf{E}} \left[ \mathbf{D}_{\mathrm{KL}}(P\|\hat{Q}(X^n)) \right] \leq \frac{k-1}{2\ln 2 \cdot n} + o(n^{-1}). \tag{12}$$

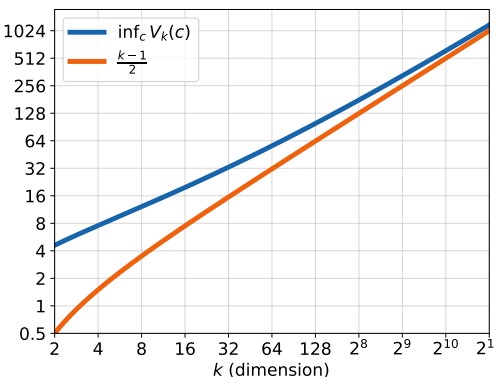 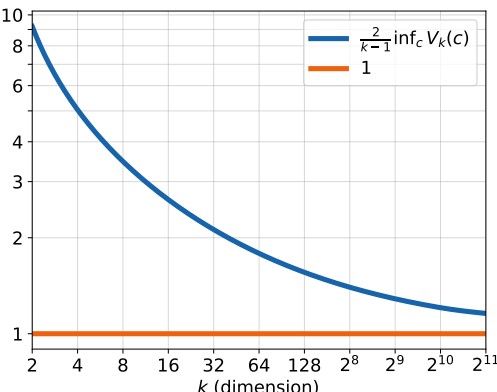

Figure 1: The optimal universal sample coding factor $\inf_c V_k(c)$, optimized over the choice of $c > 0$ for $k$-dimensional distributions, along with the lower bound factor $\frac{k-1}{2}$.

Figure 2: The optimal universal sample coding factor $\inf_c V_k(c)$, normalized by $\frac{k-1}{2}$, optimized over the choice of $c > 0$ for $k$-dimensional distributions.

## 5.2   Universal sample coding algorithm

If we could communicate a single sample with $\mathbf{D}_{\mathrm{KL}}(P\|Q)$ average bits, then alternating between sending a single sample and estimating $P$ would be the best strategy. The algorithm would take $n$ communication rounds and the total expected communication cost of this hypothetical algorithm would be:

$$\lceil \log k \rceil + \sum_{i=1}^{n-1} \frac{k-1}{2\ln 2 \cdot i} + o(i^{-1}) \leq \frac{k-1}{2} \log(n) + o(\log n), \tag{13}$$

where $\lceil \log k \rceil$ is the cost of sending the first sample. However, the real cost of sending a sample, as shown in (4), includes a constant, which would dominate the total cost as $n$ increases, making it linear instead of logarithmic. We propose a solution that requires only $O(\log n)$ communication rounds, where in each round, exponentially more samples are sent. Algorithm 1 illustrates the pseudo code for the proposed solution. The proof of Theorem 5.1 is presented in Appendix B. We first upper bound the number of bits communicated in each round. Then we show that the total number of bits communicated over all rounds is $V_k(c) \log(n) + o(\log n)$, where parameter $c$ controls the size of the groups communicated at each round. The optimal choice of $c$ depends on $k$.

We have plotted the infimum of $V_k(c)$ for different values of $k$ in Figure 1. The value of the first term of $V_k(c)$, that is $\frac{c}{\ln(1+c)} \frac{k-1}{2}$, is minimized as $c$ approaches 0 and converges to $\frac{k-1}{2}$, which is equal to the optimal redundancy factor. As we show in the next section, it is also the lower bound for any universal sample communication algorithm. The ratio between the optimal value of $V_k(c)$ and $\frac{k-1}{2}$ is shown in Figure 2, where it starts at around 9 and converges to 1 as $k$ grows; this is the multiplicative gap between the upper bound on the communication cost of the proposed algorithm and the lower bound presented in the next section.

## 5.3   Lower bound

The connection between universal source coding and universal sample coding is highlighted in the following theorem, where the lower bound of redundancy of universal source coding is the same as the communication cost of universal sample coding. This is not a coincidence, we can prove this theorem using exactly the same steps as the proof of optimality for the redundancy of universal source coding in Davisson et al. (1981).

**Theorem 5.3** (Universal sample coding - lower bound). *For any universal sample coding algorithm (Theorem 5.1), there exists $k$-dimensional distribution $P$, such that, the expected number of bits $L(n)$ required to communicate $n$ samples from $P$ satisfies:*

$$L(n) \geq \frac{k-1}{2} \log(n) + O(1) . \tag{14}$$

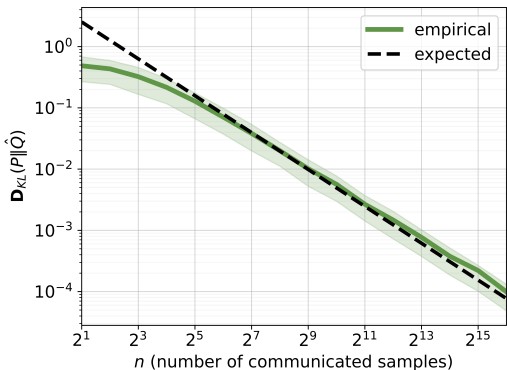

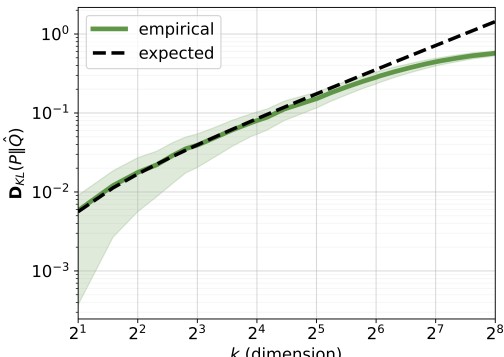

Figure 3: KL-divergence between the true and estimated probabilities for dimension $k = 8$, for a range of number of communicated samples $n$. Solid line indicates the mean, while the shaded area shows the 20th to 80th percentiles.

Figure 4: KL-divergence between true and estimated probabilities for $n = 128$ samples, for different number of dimensions $k$. Solid line indicates the mean, while shaded area 20th to 80th percentiles.

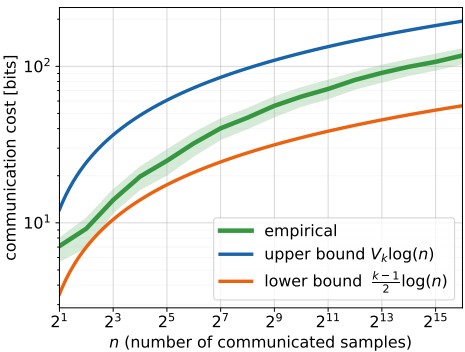

Figure 5: Communication cost of communicating $n$ samples from an 8-dimensional distribution ($k = 8$). Solid line indicates the mean, while shaded area shows the 20th to 80th percentiles.

*Proof.* Theorem 5.3 states that the claim holds for some distribution $P$, thus it holds for the supremum over all $k$-dimensional distributions $\sup_P L(n)$. We will consider $P$ as a random variable distributed according to $\Omega$. Let $\Omega$ be any distribution of $k$-dimensional distributions, then

$$\sup_P L(n) \geq \mathop{\mathbf{E}}_{P \sim \Omega}[L(n)], \tag{15}$$

since the maximum is always greater than or equal to the average. The right hand side in Equation (15) corresponds to the expected number of bits required to communicate a sample $X^n$ from $P^{\otimes n}$, where $P$ is sampled from $\Omega$. This is exactly the reverse channel coding problem, where the lower bound on the communication cost was shown in Harsha et al. (2010) to be the mutual information between the two random variables

$$\mathop{\mathbf{E}}_{P \sim \Omega} L(n) \geq I(P, X^n). \tag{16}$$

This quantity was bounded in Davisson et al. (1981) as

$$I(P, X^n) \geq \frac{k-1}{2} \log(n) + O(1). \tag{17}$$

$\square$

To show (17), Davisson et al. (1981) considers a Markov chain $P \to X^n \to \hat{Q}$, where $\hat{Q}$ is a reconstruction of $P$ based on samples $X^n$. By data processing inequality, this Markov chain implies $I(P, X^n) \geq I(P, \hat{Q})$. The quantity $I(P, \hat{Q})$ describes the minimum amount of information needed to compress $P$ in lossy source coding, and can be bounded using Shannon lower bound Shannon (1959). The final lower bound is obtained by choosing an appropriate distribution $\Omega$ and a distortion measure between distributions $P$ and $\hat{Q}$.

# 6 Empirical evaluation

To corroborate the claims about universal sample coding made in Theorem 5.1, we compare the KL-divergence and total communication cost between the theory and numerical evaluations for different number of samples and dimensions. Each experiment is repeated 1000 times, where a probability distribution $P$ is sampled from a $k$-dimensional Dirichlet distribution with concentration parameters $\alpha \in \mathbb{R}^k, \alpha_i = 1$. Then, $n$ samples are communicated using universal sample coding (Algorithm 1). We consider ordered random coding (Theis and Yosri, 2022), shown in Appendix C, as the underlying sample communication method (line 7, Algorithm 1). In Figure 3, we observe that the measured KL-divergence $\mathbf{D}_{\mathrm{KL}}(P\|\hat{Q})$ between the true probability $P$ and the estimate $\hat{Q}$ from the estimator in Lemma 5.2 follows the predicted values across a range of communicated samples. In Figure 4, we fix the number of samples to $n = 128$, but vary the dimension $k$, and show that the KL-divergence lies below the bound from Lemma 5.2. In Figure 5, we plot the total communication cost for various number of samples, which is contained between the asymptotic upper and lower bounds, $V_k(c) \log(n)$ (Theorem 5.1) and $\frac{k-1}{2} \log(n)$ (Theorem 5.3), respectively.

# 7 Federated learning (FL)

To show the efficacy of the universal sample coding in practice, we first apply it to the Federated Probabilistic Mask Training (FedPM) algorithm proposed in Isik et al. (2024). In FedPM, the weights of the neural network $w \in \mathbb{R}^M$ are randomly initialized, fixed, and known both to the clients and the central server. Each weight $w_i$ has an associated probability $\theta_i$ of that weight being masked. The training of the network consists of finding the suitable mask probabilities $\theta \in [0, 1]^M$ for all the weights using gradient descent. To test the network, a binary mask is sampled, and the effective weights of the network become $w_i * \mathrm{Bern}(\theta_i)$. In a single learning round, the server broadcasts the global mask $\theta$ to each client, which then trains an updated version $\theta'$ using its local data and communicates a sample $\mathrm{Bern}(\theta')$ from its updated mask distribution back to the central server. The new global mask probability $\theta$ is then estimated from all the received samples. The sample from $P = \theta'$ is communicated using the coding probability $Q = \theta$, which is known to both the client and the server, thereby achieving a communication cost of approximately $\mathbf{D}_{\mathrm{KL}}(\theta'\|\theta)$. To achieve a more accurate estimate of the global mask probability, we propose communicating multiple mask samples per client using universal sample coding, along with the addition of the global prior $\theta$.

For the experiments, we follow the same setup as Isik et al. (2024) using their `CONV-6` architecture on classification of CIFAR-10 images, where the data is distributed among 10 clients. A challenging scenario considered in that work is of congested clients, where only a fraction of clients contribute in each learning round. The results of our experiments are summarized in Table 2. The FL training was conducted on two Nvidia RTX 3090 GPUs, each with 25 GB of memory, with each experiment taking approximately 12 hours. The training, including preliminary experiments, took 30 days of gpu-time. If all the clients participate in each learning round the final test accuracy is around $80\%$. However, when only 1 out of 10 clients participate in the learning round the accuracy decreases to $75\%$. By communicating multiple samples in each round, we can achieve test accuracy close to the fully uncongested case. However, as argued in this work, sending multiple samples from the same distribution can be made more communication efficient by estimating the true probability from previous samples. Indeed, we observe a $37\%$ reduction in communication cost by employing universal sample coding with prior $\theta$, compared to only using the prior $\theta$.

To incorporate the prior $\theta$ into the proposed coding scheme, we use a Bayesian estimator. Although the mask is binary, we describe a general case with $k$ possible outcomes. Let $\omega \in \mathcal{P}_k$ be the prior probability, and $\omega'$ be a random variable distributed according to the Dirichlet distribution with parameters $\alpha \in \mathbb{R}^k$, for $j \in \{0, \ldots, k-1\}$: $\alpha_j = \mu \omega_j + \sum_{l=1}^G \mathbf{1}\{l\text{-th sample} = j\}$, where $\mathbf{1}\{\cdot\}$ is the indicator function, $G$ is the number of samples communicated so far, and $\mu \in \mathbb{R}^+$ is a hyperparameter controlling the reliance on the prior. The optimal coding probability $Q = \mathbf{E}_{\omega' \sim P_{\omega'}} \omega'$, i.e., for $j \in \{0, \ldots, k-1\}$, $Q_j = \frac{\alpha_j}{\sum \alpha_i}$, replaces the one in line 5 in Algorithm 1. For the binary mask case, where $k = 2$, we have $\omega = [1 - \theta, \theta]$, and $\theta'$ is characterized by a 2-dimensional Dirichlet distribution (or equivalently, a Beta distribution).

Table 2: Accuracy and communication cost of FedPM for different simulation scenarios. Values are averaged over 20 runs, with standard deviation bellow 0.003.

| FL scheme | FedPM | FedPM | FedPM | FedPM w. USC |
|---|---|---|---|---|
| # clients per round | 10 | 1 | 1 | 1 |
| # samples per client | 1 | 1 | 7 | 7 |
| final test accuracy | 0.8025 | 0.7516 | 0.8028 | 0.8039 |
| # bits per parameter | 0.6058 | 0.0340 | 0.3955 | 0.2482 |
| #bits per parameter & client & sample | 0.0606 | 0.0340 | 0.0565 | 0.0355 |

Table 3: Per token cost of sending samples from 13B model. Entropy is a lower bound for source coding, while the KL-divergence serves as a bound for sample communication.

| | # bits per token |
|---|---|
| plain text | 15.617 |
| $\hat{\mathbf{H}}(P_{13B})$ | 4.315 |
| $\hat{\mathbf{D}}_{\mathrm{KL}}(P_{13B}\|Q_{125M})$ | 1.014 |
| $\hat{\mathbf{D}}_{\mathrm{KL}}(P_{13B}\|Q_{350M})$ | 0.824 |
| $\hat{\mathbf{D}}_{\mathrm{KL}}(P_{13B}\|Q_{1.3B})$ | 0.420 |
| $\hat{\mathbf{D}}_{\mathrm{KL}}(P_{13B}\|Q_{2.7B})$ | 0.330 |
| $\hat{\mathbf{D}}_{\mathrm{KL}}(P_{13B}\|Q_{6.7B})$ | 0.266 |

# 8    Limitations and further work

Both the lower and upper bounds of communication rate (Theorems 5.1, 5.3) include a factor $k$—the cardinality of the sample space. Consequently, universal sample coding cannot be directly applied to continuous random variables, as $k$ goes to infinity so does the required communication. Universal sample coding consists of two main components: sample communication and probability estimation. While sample communication can be applied to continuous distributions, the estimation component fails because no finite number of samples can fully specify an unrestricted continuous distribution.

We propose two potential avenues for extending universal sample coding for continuous variables. Firstly, by imposing additional assumptions on the probability distribution, we could substitute counting-based estimation with a Bayesian approach. The reference distribution $Q$ would be the posterior, updated continually with each new sample. Second, we could explore a model-based approach where a common model describes $Q$. This model would be incrementally fine-tuned with the communicated samples to more accurately reflect their underlying distribution. As evidence for the potential of this idea, we apply sample communication for generating samples from a large language model on a server, using a smaller reference model at the client. While the space of text is not continuous, it is too large to estimate using count-based methods. The experimental setup and further discussion are detailed in Appendix D. As demonstrated in Table 3, the application of sample communication results in a 4- to 16-fold reduction in communication rate, depending on the auxiliary model employed. This motivates further exploration into model fine-tuning effects. Although this research currently focuses on text generation, future work will aim to extend these results to image and video generation, which represent a substantial portion of network traffic. Furthermore, determining the optimal use of new samples in the learning process presents a compelling challenge, potentially linking to advancements in online learning methods.

The universal sample communication problem, introduced in this work, involves communicating multiple samples drawn from the same probability distribution $P$, which is unknown to the decoder. An alternative but related problem would involve generating multiple samples, each from a different conditional distribution $P_{X|Z=z}$. In such a case, the optimal coding distribution would be the marginal distribution $Q = P_X$, and the optimal per-sample communication cost would be $I(X;Z)$. In the universal setting, the decoder would not know this marginal distribution. Thus, a similar estimation and sample communication approach, as demonstrated in this work, could be applied to address this

problem. However, further analysis is required to validate the communication performance of such a scheme.

# 9 Conclusion

As AI tools become ever more prevalent, it becomes increasingly important to consider the resources they consume. This paper analyzes the communication cost of transmitting samples from probability distributions, which can be seen as sending the outcomes of generative models, and show that by leveraging the fact that many generative AI applications only seek to communicate generic samples from a distribution rather than a specific sample, we can reduce the associated communication cost significantly. In this paper, we proposed *universal sample coding* for transmitting multiple samples from a distribution, which achieves the information theoretic lower bound by up to a multiplicative constant. We also applied it to a FL framework showing a communication cost reduction of $37\%$, and to remote text generation problem using large language models, showing an up to 16-fold communication cost reduction compared to sample-wise entropy coding methods.

**Acknowledgements**

This research was supported by the United Kingdom Engineering and Physical Sciences Research Council (EPSRC) for the projects AIR (ERC Consolidator Grant, EP/X030806/1) and INFORMED-AI (EP/Y028732/1).

We would like to thank Francesco Pase for sharing the federated learning codebase of Isik et al. (2024).

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

# A   Universal Estimation

The form of the universal estimator (Braess and Sauer, 2004) is provided in Algorithm 2, where $k$ denotes the size of the alphabet and $c[i]$ represents the number of occurrences of symbol $i$ in the observed sequence. Notably, when the sequence is empty, the estimator produces a uniform distribution over all symbols.

---

**Algorithm 2** Universal Estimator

---

**Require:** $c$                                     // Count of occurrences of symbols in the sequence.

1: Initialize array $\hat{Q}$ of size $k$
2: **for** $i \in \{1, 2, \ldots, k\}$ **do**
3:     **if** $c[i] = 0$ **then**
4:         $\hat{Q}[i] \leftarrow c[i] + \frac{1}{2}$
5:     **else if** $c[i] = 1$ **then**
6:         $\hat{Q}[i] \leftarrow c[i] + 1$
7:     **else**
8:         $\hat{Q}[i] \leftarrow c[i] + \frac{3}{4}$
9:     **end if**
10: **end for**

11: $W \leftarrow \sum_{i=1}^{k} \hat{Q}[i]$                                     // Normalize the distribution.
12: **for** $i \in \{1, 2, \ldots, k\}$ **do**
13:     $\hat{Q}[i] \leftarrow \hat{Q}[i] \, / \, W$
14: **end for**
15: **return** $\hat{Q}$

---

# B   Universal Sample Coding proof

*Proof.* Theorem 5.1: Universal Sample Coding

Let $c > 0$, and $G(i)$ be the number of samples communicated up to and including round $i \geq 1$:

$$G(i) = \lceil (1 + c)^{i-1} \rceil \geq (1 + c)^{i-1}. \tag{18}$$

Let $g(i)$ be the number of samples communicated at round $i$, $g(1) = G(1) = 1$, for $i \geq 2$:

$$\begin{aligned}
g(i) &= G(i) - G(i - 1) \\
&\leq (1 + c)^{i-1} + 1 - (1 + c)^{i-2} \\
&= c(1 + c)^{i-2} + 1.
\end{aligned} \tag{19}$$

At round $i$, $g(i)$ samples will be coded jointly, using an estimate $\hat{Q}$ based on previous $G(i - 1)$ samples. The cost of sending the first sample is $\lceil \log k \rceil$. For rounds $i \geq 2$, using cost from Equation (10) (with the constant upper bounded by 5) at round $i$, with the estimator from Lemma 5.2, the

expected number of communicated bits is upper bounded by:

$$\mathbf{E}\left[\, g(i)\,\mathbf{D}_{\mathrm{KL}}[P\|\hat{Q}(X^{G(i-1)})] + \log(g(i)\,\mathbf{D}_{\mathrm{KL}}[P\|\hat{Q}(X^{G(i-1)})] + 1) + 5\right] \tag{20}$$

$$\leq g(i)\left[\frac{k-1}{G(i-1)2\ln 2} + o\left(\frac{1}{G(i-1)}\right)\right] \tag{21}$$

$$+ \log\left(g(i)\left[\frac{k-1}{G(i-1)2\ln 2} + o\left(\frac{1}{G(i-1)}\right)\right] + 1\right) + 5$$

$$\leq (c + (1+c)^{-i+2})\left(\frac{k-1}{2\ln 2} + o(1)\right) \tag{22}$$

$$+ \log\left((c + (1+c)^{-i+2})\left(\frac{k-1}{2\ln 2} + o(1)\right) + 1\right) + 5$$

$$\leq c\lceil\frac{k-1}{2\ln 2}\rceil + (1+c)^{-i+2}\left(\frac{k-1}{2\ln 2}\right) + \log\left(\frac{k-1}{2\ln 2}\right) \tag{23}$$

$$+ \log\left(c + (1+c)^{-i+2} + \frac{2\ln 2}{k-1}\right) + 5 + o(1).$$

In line (20), we used Lemma 5.2 and Jensen's inequality. In line (21), the inequality $\frac{g(i)}{G(i-1)} \leq c + (1+c)^{-i-2}$ for $i \geq 2$ is used. In $\lceil\log_{1+c}(n)\rceil + 1$ rounds, there are $G(\lceil\log_{1+c}(n)\rceil + 1) \geq n$ samples communicated, and the total expected number of communicated bits, denoted by $L(n,c)$, is given by:

$$L(n,c) = \lceil\log k\rceil + \sum_{i=2}^{\lceil\log_{1+c}(n)\rceil+1}\left[c\left(\frac{k-1}{2\ln 2}\right) + \log\left(\frac{k-1}{2\ln 2}\right) + 5 + o(1) + \right. \tag{24}$$

$$\left. + (1+c)^{-i+2}\left(\frac{k-1}{2\ln 2}\right)\log\left(c + (1+c)^{-i+2} + \frac{2\ln 2}{k-1}\right)\right]$$

$$= \lceil\log k\rceil + \lceil\log_{1+c}(n)\rceil\left[c\left(\frac{k-1}{2\ln 2}\right) + \log\left(\frac{k-1}{2\ln 2}\right) + 5 + o(1)\right] \tag{25}$$

$$+ \sum_{i=0}^{\lceil\log_{1+c}(n)\rceil-1}\left[(1+c)^{-i}\left(\frac{k-1}{2\ln 2}\right) + \log\left(c + (1+c)^{-i} + \frac{2\ln 2}{k-1}\right)\right]$$

$$\leq \frac{\log(n)}{\log(1+c)}\left[c\left(\frac{k-1}{2\ln 2}\right) + \log\left(\frac{k-1}{2\ln 2}\right) + 5\right] + \left(1 + \frac{1}{c}\right)\left(\frac{k-1}{2\ln 2}\right) \tag{26}$$

$$+ \log(n)\left(1 + \frac{\ln(1+\frac{2\ln 2}{k-1})}{\ln(1+c)}\right) + \log\left(1 + c + \frac{2\ln 2}{k-1}\right) + o(\log n)$$

$$= \frac{\log(n)}{\ln(1+c)}\left[c\left(\frac{k-1}{2}\right) + \ln\left(\frac{k-1}{2\ln 2}\right) + \ln(1 + \frac{2\ln 2}{k-1}) + 5\ln 2\right] + \log(n) + o(\log n) \tag{27}$$

$$= \log(n)\left[\frac{c}{\ln(1+c)}\left(\frac{k-1}{2}\right) + \frac{\ln\left(\frac{k-1}{2\ln 2} + 1\right) + 5\ln 2}{\ln(1+c)} + 1\right] + o(\log n) \tag{28}$$

$$= V_k(c)\log(n) + o(\log n) \tag{29}$$

where the upper bound of terms dependent on round $i$ in line (25) lemmas B.1,B.2. This proves theorem 5.1. □

**Lemma B.1.** *For* $0 < c, n \geq 1$:

$$\sum_{i=0}^{\lceil\log_{1+c}(n)\rceil-1}(1+c)^{-i} \leq \frac{1}{1 - (1+c)^{-1}} = \frac{1+c}{c} = c^{-1} + 1. \tag{30}$$

*Thus,*

$$\sum_{i=0}^{\lceil \log_{1+c}(n) \rceil - 1} (1+c)^{-i} \left( \frac{k-1}{2 \ln 2} \right) \leq \left( c^{-1} + 1 \right) \left( \frac{k-1}{2 \ln 2} \right) \tag{31}$$

**Lemma B.2.** *For* $0 < c, n \geq 1$, *and* $k \geq 2$:

$$\sum_{i=0}^{\lceil \log_{1+c}(n) \rceil - 1} \log \left( c + (1+c)^{-i} + \frac{2 \ln 2}{k-1} \right) \tag{32}$$

$$\leq \lceil \log_{1+c} n \rceil \log \left( 1 + c + \frac{2 \ln 2}{k-1} \right) \tag{33}$$

$$\leq \left( \frac{\log n}{\log(1+c)} + 1 \right) \log \left( 1 + c + \frac{2 \ln 2}{k-1} \right) \tag{34}$$

$$= \log(n) \frac{\ln \left( 1 + c + \frac{2 \ln 2}{k-1} \right)}{\ln(1+c)} + \log \left( 1 + c + \frac{2 \ln 2}{k-1} \right) \tag{35}$$

$$= \log(n) \frac{\ln(1+c) + \ln \left( 1 + \frac{2 \ln 2}{(k-1)(1+c)} \right)}{\ln(1+c)} + \log \left( 1 + c + \frac{2 \ln 2}{k-1} \right) \tag{36}$$

$$\leq \log(n) (1 + \frac{\ln \left( 1 + \frac{2 \ln 2}{k-1} \right)}{\ln(1+c)}) + \log \left( 1 + c + \frac{2 \ln 2}{k-1} \right) \tag{37}$$

## C  Channel simulation

The channel simulation method used throughout this work is ordered random coding from Theis and Yosri (2022) reproduced in Algorithm 3 for convenience.

---

**Algorithm 3** Ordered Random Coding

---

**Require:** P, Q, N
1: $t, n, s^\star \leftarrow 0, 1, \infty$
2: $w = \min_x P(x)/Q(x)$
3: **repeat**
4:     $z \leftarrow$ sample $P$
5:     $v \leftarrow N/(N - n + 1)$
6:     $s \leftarrow t \cdot P(z)/Q(z)$
7:     **if** $s < s^\star$ **then**
8:         $s^\star \leftarrow s$
9:         $n^\star \leftarrow n$
10:    **end if**
11:    $n \leftarrow n + 1$
12: **until** $s^\star \leq t \cdot w$ **or** $n > N$
13: **return** $n^\star$

---

## D  Generative models

The idea behind the universal sample communication is based on two ingredients: samples can be efficiently communicated if the decoder has access to a similar distribution, and the distribution can be estimated based on observed samples. Hence, as the receiver acquires more samples, the marginal communication cost decreases. In machine learning, many models, such as classifiers or generative models, describe (conditional) probability distributions. Thus, this recipe straightforwardly translates to such scenarios, where samples from a model can be communicated in the same way, and the estimation step can be replaced by an auxiliary model that learns based on communicated samples. Specifically, let there be a server and a client, where the client wants to use a model located at the

server - for instance an artificial neural network. The model represents a conditional probability distribution $P_{X|Z=z}$, where $z$ is the input to the model. In addition, both the client and the server have access to an auxiliary model describing $Q_{X|Z=z}$. The client sends $z$ to the server, and then a sample $x \sim P_{X|Z=z}$ generated by the server's model can be communicated back with approximately $\mathbf{D}_{\text{KL}}(P_{X|Z=z}\|Q_{X|Z=z})$ bits. Then, the data point $(z, x)$, can be used for updating the common reference model $Q_{X|Z=z}$. If models $P$ and $Q$ are of the same size, this scheme could be understood as model compression, where $P$ is communicated by using samples from it, and then reconstructed at the client as $Q$. However, if model $P$ is much bigger than $Q$, then presumably the clients' model could not fit the same distribution perfectly, and the distributions may differ significantly. As such, there is a trade-off between computation/ complexity at the client and the required communication. Additionally, the model at the server might be capable at the whole range of $z \in \mathcal{Z}$, while the client might only be interested in some smaller subset of values of $z \in \mathcal{Z}' \subseteq \mathcal{Z}$. In this case, even less powerful model $Q$ could learn good approximation of $P_{X|Z=z}$ for the desired input space.

To demonstrate the validity of the idea, we perform numerical experiments in which we compare the expected number of bits required to communicate a text sample from a large language model (LLM). Instead of the estimation/learning step, we use models of different sizes from the family of Open Pre-train Transformer (OPT) models Zhang et al. (2022) (MIT license). In particular, we use the 13 billion (13B) parameter model to represent the desired conditional probability $P$, while $Q$ is represented by one of the smaller OPT models with $\{125\text{M}, 350\text{M}, 1.3\text{B}, 2.7\text{B}, 6.7\text{B}\}$ parameters, where M denotes million, and B billion. The classical way to communicate such samples is to generate the distribution at the server, sample from it, and then encode and transmit the result - we refer to this method as source coding, and, as mentioned before, it has the minimal cost $\mathbf{H}(P)$, while the sample communication approach would have an approximate cost of $\mathbf{D}_{\text{KL}}(P\|Q)$ (per sample).

In LLMs, text is represented as a sequence of tokens, where each token is a short string of characters - for instance a phrase "nocturnal animals" would be parsed into tokens ("no", "ct", "urnal", " animals"). A LLM describes the probability of the next token conditioned on the all the previous ones. This is enough to describe any distribution over a sequence, as it can be factorized in an autoregressive way $P(X^n) = \prod_{i=1}^{n} P(X_i|X^{i-1})$. To sample from such distributions, we can sample the first element, then conditioned on it sample the second one, and so on so forth. The entropy of a sequence of $n$ random variables can be decomposed as $\mathbf{H}(P^n) = \sum_{i=1}^{n} \mathbf{H}(P_{X_i|X^{i-1}})$. Conditioned on a specific $x^t$ we can get an exact entropy $\mathbf{H}(P_{X_t|X^t=x^t})$ as the model outputs probability of each token given $x^t$. However, the OPT models use 50265 possible tokens, and thus to calculate the entropy of a sequence of length $n$, one would have to calculate the entropy of $50265^n$ different sequences which is infeasible. Instead, we estimate the entropy by sampling $M = 50000$ different random sequences of length $N = 128$ according to the model. To increase variety and simulate user prompts we condition the answers on questions from OpenbookQA dataset Mihaylov et al. (2018). Then the entropy of a single token is estimated as

$$\hat{\mathbf{H}}(P) = \frac{1}{MN} \sum_{j=1}^{M} \sum_{i=1}^{N} \mathbf{H}(P_{X_i|X^{i-1}=x_j^{i-1}}). \tag{38}$$

We can similarly estimate the KL-divergence as

$$\hat{\mathbf{D}}_{\text{KL}}(P\|Q) = \frac{1}{MN} \sum_{j=1}^{M} \sum_{i=1}^{N} \mathbf{D}_{\text{KL}}(P_{X_i|X^{i-1}=x_j^{i-1}}\|Q_{X_i|X^{i-1}=x_j^{i-1}}). \tag{39}$$

The experiments were performed on 4 Nvidia RTX A6000 GPUs with 48GB of memory and PyTorch version 2.1, totalling 10 hours of wall-clock-time.

The estimated expected communication cost per token is shown in Table 3. Even the smallest 125M model allows us to decrease the communication requirement by more than 4 times. As the auxiliary model gets bigger, the communication cost decreases by up to 16-times. These results are asymptotic, and do not include the overhead for shorter sequences. Thus, we also present upper bounds for communication of groups of B $\in \{1, 2, 4, 8, 16, 32, 64, 128\}$ tokens jointly in Figure 6. For sample communication, the upper bounds are calculated by including the logarithmic and constant terms from Equation (4). For source coding, we use the upper bound from Equation (5). The sample communication approach beats source coding even for a group size B $\geq 2$ for any of the auxiliary models. Only for B $= 1$ source coding is more efficient, as the constant overhead from equation (4)

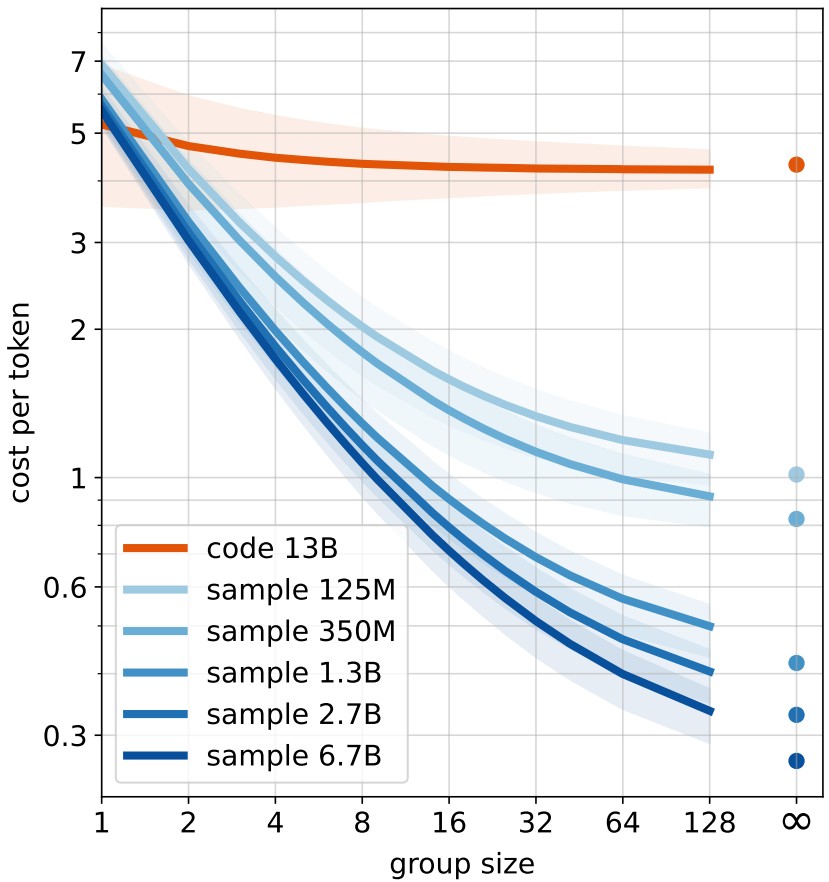

Figure 6: Communication cost per token as a function of the group size. The solid lines depict the mean, while the shaded areas correspond to the 25th to 75th percentiles. The 'code 13B' indicates the source coding approach, whereas 'sample $Z$' pertains to sample communication with the auxiliary model $Z$.

is 5, which is greater than the source coding cost. Even for small group sizes of $B = 8$, the sample communication approach achieves $2 - 4$ times reduction in the communication cost.

