# OpenReview forum: "Universal Sample Coding"
_NeurIPS.cc/2024/Conference — NeurIPS 2024 poster_

### Official Review · Reviewer_Jywa · 2024-06-12

**Soundness:** 3
**Presentation:** 3
**Contribution:** 3
**Rating:** 7
**Confidence:** 5

**Summary:**

The authors consider the problem of efficiently encoding a sequence of $n$ iid realisations $X_i \sim P$ using as few bits as possible. This scenario is different from one-shot channel simulation because it involves multiple rounds of communication. Between these rounds, the encoder and decoder adjust their coding distribution, leading to substantial improvements in compression performance. Using techniques and results from universal source coding, the authors characterise the optimal lower bound for this task and propose a scheme that achieves the lower bound within a multiplicative constant. They conduct experiments on a toy federated learning problem, showing that their technique could make federated learning significantly more robust when only a fraction of the clients participate in each communication round while reducing communication costs.

**Strengths:**

Overall, I found the paper quite enjoyable to read. The motivation and problem set-up are clear. While at a high level, the authors' solution to the problem is "the obvious one," it has many subtleties, e.g. choosing the "right" probability estimator in Section 5.1. Thus, this solution, combined with the analysis, provides interesting new insights for the efficient communication of samples. I have checked all the proofs in detail, and they are correct. I also liked the federated learning experiments, as they present a scenario in which the authors' method brings concrete benefits.

**Weaknesses:**

There are three major points that, if clarified, would make the paper significantly stronger:

1. **What is the sample complexity/run time of the authors' proposed solution?**
Since Algorithm 1 calls an arbitrary channel simulation algorithm a subroutine, it should be relatively easy to calculate the total runtime in terms of the subroutine's runtime and state it as an accompanying result to Thm 5.1. Such a result would benefit people wishing to build on the authors' work, as the general issue with channel simulation algorithms with target $P$ and proposal $Q$ is that their runtime scales with $\Vert dP/dQ \Vert_\infty \geq \exp(KL[P \Vert Q])$.
2. **Clarifying the relationship between universal source coding, universal sample coding and channel simulation.** The authors' framework has key differences compared to channel simulation as well as universal source coding, which should be illustrated better. First, they switch to a streaming/multi-round communication setting, where the encoder and decoder can update their coding distribution, which differs from one-shot channel simulation as, in the latter case, there is only a single round of communication. Second, they fix the target distribution of the elements of the sequence, which is equivalent to assuming a delta distribution on the source in channel simulation. Similarly, the analogous setting in universal source coding would be to have a delta distribution as the source: for an alphabet of $K$ symbols, we wish to encode $n$ copies of the same fixed symbol. This can be done by encoding which symbol is repeated using $\log K$ bits and then encoding the number of repeats, which can be done using $\log n + O(\log\log n)$ bits with Elias delta coding. Therefore, the authors' setting and solution are closer to a universal code (a prefix code on the integers with a certain universality property) than a universal source code (a code which doesn't need to be prefix, but the ratio of the actual and optimal expected codelengths must go to one as the number of symbols increases). I realise that this terminology is annoyingly similar. Still, pointing this out/fixing this would enhance the clarity of the paper, especially given that in the current version, the authors seem to use the terms universal code and universal source code interchangeably. Furthermore, in the same way, that a universal code can be used to create a universal source code, I believe we could use the authors' proposed method to create a closer analogue of universal source coding: Assume we have two correlated variables $X, Y \sim P_{X, Y}$ with unknown marginal $P_Y$, if Alice receives $X \sim P_X$ and wants to communicate $Y \sim P_{Y \mid X}$. We could apply the authors' method of updating the counting distribution such that the approximating distribution converges to the marginal $P_Y$ so that the scheme's expected codelength will asymptotically be $n \cdot I[X; Y]$.
3. **The authors only state results assuming a uniform initial distribution.** There are two reasons why I put this as a major issue: 1) this assumption is not explicitly stated in the paper, but the authors make use of it both in the explanations as well as the proofs. 2) At the end of section 7, the authors suggest using a different initial distribution, whose performance is thus not covered by the theoretical statements, even though this setting is of the greatest practical relevance. I would conjecture that the initial choice should only affect the results up to a constant additive factor.

I have a few more minor issues as well:

1. It would be good to report the scheme's overhead compared to the theoretical lower bound in the FL scenario, akin to Figure 5.
2. I would need some clarification on the FL experimental setup and results: how did the authors set mu in the FL scenario? Why did the authors specifically pick seven samples per client? How does the performance change as a function of the number of samples sent per client and the fraction of participating clients?
3. "We consider ordered random coding Theis and Ahmed (2022) as the underlying sample communication method" - The authors should clarify (possibly by giving pseudocode in the appendix) what exact implementation of ORC they used. I imagine that the authors did the "right thing" and used the discrete version by drawing samples without replacement, and they either ran ORC until the sample space was exhausted or used a more sophisticated early termination criterion. In any case, it would be crucial to state this, as ORC is usually regarded as an approximate sampling scheme (and is usually used for continuous distributions).

Finally, I found some typos/issues with the notation and writing:

 - line 99: "non-asymptomatic"
 - "the sample communication bound does not rely on using the exact marginal distribution P, and still holds for other reference distributions Q" - I'm not sure what this sentence means.
 - Nitpick: the authors state in section 2 that all logs are in base 2, but then they use ln (eg in eq 11)
 - Theis and Ahmed (2022)—The reference is incorrect; the second author is Noureldin Yosri, so it should be Theis and Yosri (2022).
 - "In a single learning round, the server broadcasts the global mask θ to each client, which trains an updated version θ′ using their local data and then communicates a sample Bern(θ′) from its updated mask distribution to the central server, where the new global mask probability θ is estimated from all the samples" - It is quite difficult to parse this sentence; please split it in two.
 - I believe that $\hat{P}$ (in Section 6 and Figures 3 and 4) and $\hat{Q}$ (everywhere else) mean the same thing. Please update them to be consistent if so.

**Questions:**

- Could we perhaps extend the result to continuous spaces by using KDE, or is there something fundamentally different about the continuous case?
 - "However, the real cost of sending a sample, as shown in (4), includes a constant, which would dominate the total cost as n increases, making it linear instead of logarithmic." - Does it? In the authors' case, since the approximating distribution converges to the true target, the ORC index distribution should converge to (approximately) the delta on 1. Hence, I would imagine that the overhead the authors mention would vanish if we used a universal source code to compress the ORC indices.
 - What is the long-term behaviour of the approximating distribution? My initial estimate would be that we have $O(k^{-1}n^{-1/2}) $ convergence to the target by the CLT; could we say something more precise? Is there a good way to speed up the convergence, maybe?

**Limitations:**

The authors should address the three main points I laid out in the Weaknesses section.

---

> ### Author Rebuttal · Authors · 2024-08-07
>
> **Computational Complexity:**
> As pointed out by the reviewer, the complexity of all currently known methods for exact channel simulation is proportional to $|\frac{dP}{dQ}|_\infty$. Thus, the proposed method has computational complexity
>
> $$\sum_{i=2}^{\lfloor\log_{1+c} (n)\rceil+1} \left(\left\Vert\frac{dP}{dQ(X^{G(i-1)})}\right\Vert_\infty\right)^{g(i)}.$$
> To describe the complexity in a more informative way the behavior of the ratio infinity norm between $P$ and its estimate is required.
> We do not have such results, besides some preliminary calculations suggesting that it might be bounded by $(1+\epsilon)+O(e^{-n}n^{1/\epsilon})$ in expectation if $n$ symbols are communicated.
> In practice, we used the ordered random coding algorithm with a limited number of samples, at or above $2^{D_{KL}}$. This could have negative implications for the estimation accuracy, as samples do not exactly follow the true distribution $P$. However, from our experiments, we did not observe such negative outcomes.
>
> **Naming and generalization:**
> We agree with the reviewer's perspective about the generalization of channel simulation to multiple samples, where each sample is drawn from a different distribution $P$. Then, the same counting-based estimation could be applied to achieve an optimal asymptotic rate by learning the marginal $Q$. Our formulation generalizes the classical channel simulation to generating multiple samples from the same distribution, while the proposed framework further generalizes this to varying the target distribution at each sample. This setting can be called as 'Universal Channel Simulation', which we believe is a great avenue for further research.
> On the other hand, universal source coding aims to answer the question "How to send $n$ samples from an unknown distribution $P$?," while the universal sample coding can be understood as "How to remotely generate $n$ samples from an unknown distribution $P$?". Moreover, the redundancy of universal source coding and the rate of universal sample coding coincide, in a natural way.
>
> We agree with the reviewer that the relationship between universal source coding, universal coding, channel simulation, and universal sample communication can lead to confusion. Thus, we have replaced all references to 'universal coding' with 'universal source coding'.
> We have added a comment explaining the channel simulation generalization to n-letter case proposed by the reviewer.
>
> **Prior distribution:**
> The uniform prior distribution is a consequence of the way the estimator of Lemma 5.2 is defined - assigning a weight to each symbol based on the number of its occurrences so far. Thus, without any observations, it assigns equal weights to all symbols, and consequently a uniform prior. For the federated learning experiment, we have switched this estimator to a Bayesian one since we had access to side information - the Bernoulli parameter value in the previous step. If the distribution of $P$ follows that implied by the prior this is an optimal estimator, and therefore performs better for any finite $n$ than the agnostic one from Lemma 5.2.
> Influence of such prior diminishes with samples with rate $n^{-1}$, additionally, we know that the concentration of the estimator is of order $n^{-1/2}$. Thus, asymptotically, the prior does not change the asymptotic behavior of the estimator (barring extreme cases of having infinite-weight prior). We conjecture that, the influence of such prior decreases exponentially as more samples are communicated.
>
> To clarify, we will provide the exact form of the estimator in Theorem 5.2 in the Appendix, and state in the main text that the default prior for the first sample is uniform. Additionally, we changed the wording in the federated learning section to highlight that a Bayesian estimator is used.
>
> **Federated learning:**
> For the federated learning experiment, we have run an experiment with $10, 4, 2$, and $1$ active clients, and tested the effects of sending different number of samples (up to around 16). As expected, the performance was robust to the choice of $\mu$. For the experiments we used $\mu=4$, but $\mu\in[3,10]$ resulted in a similar communication cost, while larger values were converging to the cost of independently sending all the samples. Due to time limitations, we were unable to provide the gap to optimality for the federated learning scenario in this rebuttal.
>
> **Questions:**
> 1. Yes, extension to the continuous case using kernel density estimation is possible. We are unaware of the convergence rate of such methods, but it would be an interesting future direction.
> 2. By describing the cost of sending each sample separately as linear, we were referring to the upper bound on the communication rate implied by Poisson functional representation. However, we admit that if the samples are sent jointly this cost could be sublinear.
> 3. Our understanding is that the estimator does indeed converge at the rate dictated by the central limit theorem. We are unaware of any more precise statements or ways to speed up convergence.
>
> We agree with all the other issues pointed out by the reviewer and will revise the manuscript accordingly.

---

> > ### Comment · Reviewer_Jywa · 2024-08-10
> >
> > I thank the authors for their response. They addressed my concerns, and I increased my score accordingly.

---

### Official Review · Reviewer_NNh9 · 2024-06-12

**Soundness:** 3
**Presentation:** 3
**Contribution:** 4
**Rating:** 7
**Confidence:** 4

**Summary:**

This paper introduces Universal Sample Coding. This is a simple but significant extension to channel simulation where the sender and receiver communicate $N$ ($N>>1$) samples. The authors prove that the expected codelength per sample will be negligible when $N\rightarrow \infty$, and verify with toy examples. The authors also demonstrate the application of this coding scheme in federated learning and the communication of generated samples from generative models.

**Strengths:**

1.  The author formulated the universal sample coding problem and proposed a practical method to handle it. The idea of extending channel simulation to multiple samples is simple yet can have significant influence, and the methods described in this paper are practical and directly applicable to (perhaps small-scale) discrete distributions.
2. The authors also demonstrate a suitable application of the proposed method: in federated learning and the communication of generated samples from generative models. I suspect that in practice, we will want to communicate so many samples to ensure the per-sample cost vanishes in the latter scenario, but this idea is neat and supportive of the proposed method itself.

**Weaknesses:**

My major concern with this method is the runtime.
To ensure the per-sample cost is asymptotically 0, the authors propose a solution in which exponentially more samples are sent in each communication round. This means that when the problem scale is large, the $D_{KL}[P||Q]$ in some rounds can be large.
For order random coding the author used in this paper, to achieve a low biased sample, the total sample size will be $O(2^{D_{KL}[P||Q]})$.
$D_{KL}[P||Q]$ will, in the end, go to 0, but it will happen only after enough samples are sent.
Therefore, I kindly ask the author to provide the sample size and the KL divergence for each problem and each round, which can provide more hints on how well this method can scale.



This is overall a good paper and I will be happy to further raise my score if the questions and weakness are addressed.

**Questions:**

Besides Weaknesses, I have one further question:
Lemma 2 says an estimator exists for any distribution but does not provide the form for this estimator.
Do you use this estimator for all of the experiments? In Fig 5, I can observe that the empirical results in this toy example obey the bounds very well. But do you think this will always be the case for the FL and LLM settings? Is there any potential caveat?

**Limitations:**

adequately addressed

---

> ### Author Rebuttal · Authors · 2024-08-07
>
> **Complexity:**
> To answer the reviewer's question about the occurrence of outliers in KL divergence between P and its estimate we have plotted it for every round of sending $n=2^{14}$ samples from $k\in\{3,8\}$-dimensional distribution $P$. In every run $P$ is sampled from a Dirichlet distribution parameterized by $\alpha=(1,\dots,1)$. The plot is appended in the global rebuttal.  The blue dots show the KL-divergence in every round for every run and red line shows the mean KL-divergence. Yellow lines show proportion of total runs at or below specified value of KL divergence, that is every 10-th percentile, as well as 95th and 98th percentiles, while black line shows the upper bound on expected KL (ignoring some minor terms). We indicate the number of samples sent in each round below the x-axis.
> We can see that most of the samples concentrate around the mean. KL values up to 20 are reasonable, which is more than $90$\% of the time, and up to 30 is still computationally feasible (above $98$\%). For the few outliers we can either accept the high bias samples or incorporate a simple feedback mechanism - if the KL does not decrease enough, keep the number of samples the same as in the previous round. This marginally increases the rate by adding one bit of information for every round, totaling $\log(n)$ bits.
>
> **Estimator:**
> The estimator in Theorem 5.2 assigns a weight to each symbol based on the number of occurrences observed in the samples communicated so far plus some small bias, and the estimated probabilities are obtained by normalizing these weights. The exact form of this estimator will be included in the Appendix. The importance of this estimator is that it achieves a min-max lower bound in KL divergence between any distribution $P$ and its estimate.
> We use this optimal 'P-agnostic' estimator for the numerical experiments, except in the federated learning example, where we use a Bayesian estimator, which works in the same way except for the bias constants.
> Knowing that the Bernoulli parameters -- by which the neural network is characterized --should not change significantly between training rounds we bias the initial estimate / prior ($Q$) to be close to its value before the training epoch. We cannot claim the optimality of this estimator for an arbitrary $P$, but since we know that in reality $P$ will be correlated to its previous value we see a reduction in the number of communicated bits.
> In the LLM example, the state space is too large to be estimated directly with count-based methods. In that scenario, we would like to investigate fine-tuning a small language model or potentially conditioning one on all text samples communicated so far. This is left for further study.

---

> > ### Comment · Reviewer_NNh9 · 2024-08-08
> >
> > Thank you for your reply and your additional results on the KL. The plot clear and I recommend including this plot in your paper later.
> >
> > I agree that KL values up to 20 are reasonable. However, I do not necessarily agree that KL up to 30 (ideally, at least 2^30 samples if your KL is in bits) is still computationally feasible, especially considering the setting of FL where each client should not have very strong computational resources.  Could you explain how many samples you use for this KL value?
> >
> > >  For the few outliers, we can either accept the high-bias samples or incorporate a simple feedback mechanism
> >
> > What did you do in your experiments? If you just accept the high bias, how does this bias influence, e.g., convergence?

---

> > > ### Author Response · Authors · 2024-08-09
> > >
> > > For the experiments, we draw $2^{\min(D_{KL}, 25)}$ samples per round. The high KL is caused by the large number of samples sent which is not the case in FL. We agree that this might be an issue for the generative case, however, in those settings greater computational resources are usually available.
> > >
> > > We think that using biased samples can delay or prevent convergence the convergence. Empirically, in the data provided in the global response (excluding the last round with fewer samples), the probability that: given KL>25 for i-th round, KL>25 for (i+1)-th round is 37%. Thus in most of the cases,  the estimate goes 'on the right track' after diverging.

---

> > > > ### Comment · Reviewer_NNh9 · 2024-08-09
> > > >
> > > > > Empirically, in the data provided in the global response (excluding the last round with fewer samples), the probability that: given KL>25 for i-th round, KL>25 for (i+1)-th round is 37%. Thus in most of the cases, the estimate goes 'on the right track' after diverging.
> > > >
> > > > Thank you for your reply. I think showing these results in some way (for example, adding an extra plot or tab) can make your paper stronger.
> > > >
> > > > I feel my concerns have been addressed. Thus, I raise my rating from 6 to 7.

---

### Official Review · Reviewer_YZzm · 2024-06-24

**Soundness:** 2
**Presentation:** 3
**Contribution:** 3
**Rating:** 5
**Confidence:** 4

**Summary:**

This paper studies the problem of communicating multiple samples from an unknown distribution using as few bits as possible. The authors provide upper and lower bounds on its communication cost that are within a multiplicative factor from each other. The upper bound is derived from analysing the communication cost of an universal sample coding algorithm, which the authors design based on the reverse channel coding results in information theory. The lower bound is derived based on analysing the connection between universal source coding and universal sample coding. Experiments show that the universal sample coding algorithm can reduce the communication cost in a Federated Learning scenario (up to $37\%$) or for generating samples from a Large Language Model (LLM) on a server (up to $16$ times).

**Strengths:**

+ The problem setting looks interesting.
+  Experiment shows that communicating multiple samples can improve the test accuracy (Table 2).

**Weaknesses:**

+ The gap between lower and upper bounds (the ratio between $\inf_c V_k(c)$ and $(k-1)/2$) is quite large for not very large $k$ (See Fig. 1 and Fig. 2).
+  Algorithm 1 requires a shared source of randomness between the encoder and decoder.

**Questions:**

+ Please address the weakness comments above. Please explain how to generate the share source of randomness in practice.
+ What is the distribution of the random string $Z \in \mathcal{Z}=\\{0,1\\}^{\infty}$ in Theorem 5.1?
+ What are the definitions of $I(P,X^n)$ and $I(P,\hat{P})$ in (16)?

**Limitations:**

This is a theoretical research paper, hence the negative society impact of this work is not direct. The authors mention some technical limitations of this work in Section 8.

---

> ### Author Rebuttal · Authors · 2024-08-07
>
> The gap between the upper and lower bounds quickly diminishes as the dimension increases. In our approach, we focus on minimizing the upper bound by choosing an appropriate constant $c$. However, a different $c$ might work better empirically, although lacking optimality in the derived upper bounds. Federated learning experiments validate the competitiveness of the proposed scheme in practice.
>
> From a mathematical perspective, common randomness is an infinite list of fair coin flips - that is $Z_i\sim$ Bernoulli(0.5). Equivalently it can be understood as a binary expansion of a single uniform random variable from interval $[0,1)$. This shared randomness can then be used to draw a common list of random variables characterized by $Q$ using, for instance, inverse transform sampling.
> In practice, it is sufficient that the encoder and decoder have a common seed combined with pseudo-random number generation, which is then used to draw common samples in tandem. Availability of unlimited common randomness is a standard assumption in the channel simulation literature.
>
> The reviewer was rightfully confused with our abuse of notation in the mutual information expressions. Here, we will write it more explicitly to dispel any confusion. Let $\Omega$ denote the set of all $k$-dimensional discrete distributions (i.e., distributions over a discrete alphabet of size $k$). Let $\Pi$ denote a random variable taking values in set $\Omega$. Here, $P$ denotes the particular realization of $\Pi$.
>
> Theorem 5.3 states that the claim holds for some distribution $P$; thus, it holds for the supremum over all distributions in $\Omega$. Then,
>
> $$\sup_{P \in \Omega} L(n) \geq \mathbf{E}_{\Pi}[L(n)],$$
>
> since the maximum is always greater than or equal to the average. Note that $\Pi$ and $X^n$ are correlated random variables, and the right hand side of Equation (15) corresponds to the expected number of bits required to communicate a sample $X^n$ conditioned on $\Pi = P$. This is exactly the reverse channel coding problem, where the lower bound on the communication cost was shown in Harsha et al. (2010) to be the mutual information between the two random variables
> $$\mathbf{E}_{\Pi} L(n) \geq I(\Pi; X^n).$$
> This quantity was bounded in Davisson et al. (1981) as
> $$I(\Pi; X^n) \geq \frac{k-1}{2}\log(n)+O(1).$$

---

> > ### Comment · Reviewer_YZzm · 2024-08-14
> > **Reply to the authors' rebuttal**
> >
> > Thank you very much for your answer to my questions. However, based on your answers, I keep my score unchanged.

---

### Official Review · Reviewer_RwEb · 2024-07-20

**Soundness:** 3
**Presentation:** 2
**Contribution:** 3
**Rating:** 5
**Confidence:** 2

**Summary:**

The paper proposes a new problem called "Universal Sample Coding". This is related to a problem called reverse channel coding (or channel simulation) where the receiver must generate samples from a target distribution (P) that is known to the sender but not the receiver. In addition, the receiver and sender have shared randomness which is used to generate samples from a reference distribution Q. The goal is to characterize the communication complexity i.e., the number of information bits that must be transmitted from the sender to the receiver to generate the target sample(s).

As far as I understand the authors propose a new problem setting where  (at least) the decoder also does not know the reference distribution Q, but estimates it. The paper provides upper and lower bounds on the communication complexity in this variant that have the same scaling in terms of the number of target samples.

**Strengths:**

Please see the summary above. Although the setup considered in the paper is potentially new, I did not fully understand it.

**Weaknesses:**

The setup and results need clarification.

1. Can you clarify your problem setup? Is it that both the encoder and decoder do not know the samples?  Who generates the samples $X_1, \ldots, X_n$ generated from the reference distribution and how are they known to the encoder/decoder?   In the reverse channel coding/channel simulation settings that I am familiar with, both the sender and receiver know the reference distribution $Q$ and use common randomness to generate identical samples.  I wasn't sure how the samples are generated.

2. The statement of Thm 5.1 is not clear to me. Usually the rate bound should be on $H(f(P,Z)|Z)$  as $Z$ is known to both the encoder and decoder. Why do you consider $H(f(P,Z))$ as your rate?

3.Theorem 5.1 is for "some c", but in the numerical bounds you are optimizing over $c$. Should the statement be for every $c$?

**Questions:**

please see the weakness section above..

**Limitations:**

yes

---

> ### Author Rebuttal · Authors · 2024-08-07
>
> Our problem setting is indeed very similar to conventional channel simulation as described by the reviewer, with the following difference: instead of a single sample, the goal of the decoder is to generate multiple samples from the target distribution $P$, which is known only to the encoder. To be precise, our goal is to identify the average number of bits that need to be communicated to the decoder so that it can generate $n$ samples from $P$. Our solution relies on the idea that as the decoder generates samples from $P$, it can also estimate it, and its estimate can be used as the reference distribution $Q$ for the remaining samples. As the decoder generates more samples, it will have a better and better estimate of $P$, and hence, it will cost fewer and fewer bits to generate new samples from $P$. To the best of our knowledge, this is a new problem formulation that was not studied before and is also relevant for many practical scenarios, as we argue in the paper.
>
> We thank the reviewer for pointing out the two typos. Theorem 5.1 should read as $H(f(P,Z)|Z)$, and does indeed hold for `any $c$'. We have corrected both in the manuscript.

---

> > ### Comment · Reviewer_RwEb · 2024-08-09
> > **Follow up question on problem formulation**
> >
> > Thank you for clarifying the problem formulation. I was wondering why is the problem setup not already a special case of the one-shot setup in channel simulation?
> >
> > Here is one specific example of one-shot setting: The encoder observes $X \sim p_X(\cdot)$ and the decoder wants to sample $Y \sim p_{Y|X}(\cdot)$. We know from prior works that the communication rate required is approximately $I(X;Y)$. In your setting you want to sample $Y_1,\ldots, Y_n$ i.i.d from $p_{Y|X}(\cdot)$ then using their scheme and defining $Z=(Y_1, \ldots, Y_n)$ the rate of  $I(X;Y_1, \ldots, Y_n)$ is naturally achievable.
> >
> > Even if you don't assume $X$ to be random, a natural connection to one-shot schemes can be made in a similar fashion. Can you provide a comparison between your proposed approach a natural extension of one-shot schemes?

---

> > > ### Author Response · Authors · 2024-08-10
> > >
> > > The reviewer is right. In fact, this is what we state on page 4 of the paper to derive equation (10). One can directly use any standard channel simulation technique to directly generate $n$ samples. (In the conditional version of the problem suggested by the reviewer, this would be equivalent to the mutual information expression). Our work uses this fact for min-max analysis i.e. guarantees for any distribution $P$.

---

> ### Comment · Reviewer_RwEb · 2024-08-12
>
> Thank you for helping me better understand your contributions. I will keep my score as it is.

---

### Author Rebuttal · Authors · 2024-08-07

We would like to thank all the reviewers for their feedback, most of our rebuttals are specific to the reviews, and thus we respond to each individually.


To answer the question by reviewer NNh9, we plot the empirical KL-divergence.

---

### Decision · Program_Chairs · 2024-09-25

**Decision:**

Accept (poster)

**Comment:**

This paper makes a significant advancement on the classical channel simulation problem in which the reference distribution $Q$ is only known via samples. The authors derive information-theoretic bounds on the expected communication cost and also derive a practical algorithm that achieves the lower bound up to a multiplicative constant. Applications to federated learning are presented. This paper makes significant contributions and is enjoyable to read. I believe it will be of interest to the NeurIPS audience.